# Influence of Climate Warming on Grapevine (*Vitis vinifera* L.) Phenology in Conditions of Central Europe (Slovakia)

**DOI:** 10.3390/plants10051020

**Published:** 2021-05-20

**Authors:** Slavko Bernáth, Oleg Paulen, Bernard Šiška, Zuzana Kusá, František Tóth

**Affiliations:** 1Department of Fruit Production, Viticulture and Enology, Faculty of Horticulture and Landscape Engineering, Slovak University of Agriculture, Tulipánová 7, 94976 Nitra, Slovakia; slavko.bernath@uniag.sk (S.B.); zuzana.kusa@gmail.com (Z.K.); 2Štúrova 42, 94901 Nitra, Slovakia; bbeetless@gmail.com; 3Gemerprodukt Valice-OVD, Okružná 3771, 97901 Rimavská Sobota, Slovakia; ftoth@gemerprodukt.sk

**Keywords:** climate change, grapevine, phenophase

## Abstract

The impact of warming on the phenology of grapevine (*Vitis vinifera* L.) in conditions of central Europe was evaluated at the locality of Dolné Plachtince in the Slovakian wine region. In Welschriesling and Pinot Blanc model varieties, the onset of phenophases as defined in the BBCH scale over the period of 1985 to 2018 was observed. Based on the data obtained, the influence of the average and average maximum temperature and GDD on the onset of phenophases was evaluated. The results observed indicate earlier budburst by five to seven days, earlier beginning of flowering by 7 to 10 days, earlier berry softening by 18 days, and harvest dates advanced by 8 to 10 days on average. In both varieties, the highest influence of the average monthly temperature in March on budburst, the highest influence of the average monthly temperature and the average maximum temperature in May on the beginning of flowering, and the highest statistically significant influence of the average maximum temperature in June on the softening of berries was found. Warming observed in moderate climate conditions of northern wine regions in central Europe (Slovakia) has not yet caused changes in the grapevine phenology stable enough to require serious adaptation measures.

## 1. Introduction

In many regions of the world, various phenomena have been observed that are attributed to climate change. Warming of an environment is among the basic phenomena related to climate change. Observed changes in 27 premium wine regions across the globe have shown an increase of the average growing season temperature from 1950 to 2000 by 1.3 °C [1], while in Europe, the increase was 1.7 °C from 1950 to 2004 [2,3,4]. However, the rate of warming that has been observed over the recent decades varies among wine regions in Europe. On the Adriatic coast of central Italy, the average annual temperature increased by more than 0.5 °C from the 1970s [5], though the warming rate has weakened during the last decade. In the Palatine region of south-western Germany, the increase was 2.1 °C from the 1970s [6]; in the department of Hérault in Mediterranean France, it was 1.3 °C within the 1980 to 2006 period [7]. From 1991 to 2014, the average annual temperature increased by 0.9 to 1.1 °C compared to its average value in the 1960 to 1990 period on Slovakia′s territory [8]. These data suggest great variability of climate change manifestation among different regions, which should be considered.

The impact of climate change is manifested not only in tendentious changes but also in the occurance of extreme events (warm winters, spring frosts, etc.) with increasing frequency and seriousness [9]. This justifies the attention paid by the viticultural sector to climate change. Grape production is higly sensitive to climate change and the key channels of its influence on grape and wine are through temperature, water, weather extremes and possibly CO_2_ itself [10]. The possible strong stimulative effect of rising atmospheric CO_2_ is reported [11].

According to the HadCM3 climate model average, predicted temperatures for the high-quality wine producing regions will increase by 2.04 °C within the 2000 to 2049 period [3]. Projected changing climate will have a significant effect on the European viticultural geography. It will have detrimental impacts in southern Europe, mainly due to increased dryness and cumulative thermal effects, while regions in western and central Europe will benefit with higher wine quality and emerging new potential areas for viticulture [12,13,14].

Climate change influences agriculture/viticulture and calls forth adaptation and mitigation actions for farming communities to be more resilient. Viticulture is among the most challenged sectors. Understanding the changing suitability of regions for viticulture under climate change will help to develop adaptation strategies in traditional winegrowing regions [15]. In order to maintain profitability and to ensure long-term future, producers will be required to adapt to changing climatic characteristics. Planning for adaptation is complicated due to uncertainty about future conditions; management strategies are more often influenced by more pressing immediate concerns [16].

Climate change is exerting an increasingly profound influence on vine phenology and grape composition, and ultimately affects vinifications, wine microbiology and chemistry, and sensory aspects. [17]. The phenology of grapevine and berry ripening indices are extremely sensitive to the climate and highly genotype-specific. Climate characterization and monitoring of grapevine phenology and berry biochemistry are efficient tools to define the environmental vulnerability of wine regions and create a basis for addressing strategies for future planning of viticulture practices [18]. Temperature is an essential factor influencing the duration of phenophases. Meteorological conditions, including air temperature, are changing from year to year, causing variations in the onset and length of phenophases as well as grape quality [19]. In various studies, a diversity in the phenological response of grapevine cultivars was recorded with potential utilisation in adaptation to climate change. Later ripening cultivars advanced faster than earlier ripening ones [20]. Variability of the phenological response has been observed not only between distant regions but also over short distances in a wine region, which is related to local characteristics [21]. There are different phenological responses between vicinal regions with different altitudes. In Georgia, for example, from 1994, the advance of Rkatsiteli cv veraison was 5.9 days for 250 to 500 m a.s.l., while it was 18.1 days for 750 to 1000 m a.s.l. [22].

In established winegrowing regions, over a long history, growers have optimized yield and quality by choosing plant material and viticultural techniques according to climatic conditions, but as the climate changes adaptation measures that include changing of plant material (cultivars, rootstocks) [23,24] and viticultural techniques, e.g., changing trunk area, leaf area to yield ratio, pruning time, are inevitable to maintain the optimal period of harvest dates [25].

Climate plays a vital role in the terroir of a given wine region as it controls the factors that determine wine attributes and typicity. Changes in viticultural suitability over the last decades, for viticulture in general and for specific cultivars, have been reported for many wine regions. These will reshape the geographical distribution of wine regions and wine typicity may also be threatened in many cases [26]. Warmer climate conditions might be beneficial for maintaining the current grape cultivars with their current grape quality or cultivating new grape cultivars in new projected winegrowing areas [27]. Results of the study performed in Romania revealed a 2.4 million ha expansion of the area with the climate suitable for wine production within the country, a 180 m increase in altitude with the suitable climate up to a current maximum of 835 m a.s.l., and northward shift of 0.036° of area with the suitable climate [28]. In Hungary, a significant change of growing degree days (GDD) has been observed over the past 30 years—by 1240 °C—and the total area planted with the cultivars that need more warmth has been extended [29]. Growing of grape has been spreading to new areas during recent decades. Warming of the climate has been accelerated in the last few decades and has made it possible to grow some hybrid vine varietes and even some *Vitis vinifera* cultivars even in South Finland (Helsinki region) [30].

## 2. Results

### 2.1. Temperature Data

The mean annual temperature ranged from 8.7 °C in 1996 to 11.55 °C in 2014 (Figure 1, Appendix A). Over the last 10 years assessed (2009 to 2018), it was on average 10.5 °C, representing an increase of 1.4 °C compared to the long-term normal between 1960 and 1990. Trend analysis of the mean annual temperature over the assessed period shows a statistically significant increase (*r* = 0.6, *p* = 0.0002).

The average temperature of the growing season (IV. to IX.) over the evaluated 1985 to 2018 period was from 15.6 °C in 1991 to 18.5 °C in 2018 (Figure 1). Over the last 10 years assessed, it reached 17.5 °C, representing an increase of 1.5 °C compared to the long-term normal of the 1960 to 1990 period. Trend analysis shows a statistically significant increase of the mean growing season temperature (r = 0.57, *p* = 0.0004).

When assessing the changes of the average monthly temperatures, we compared the average temperatures over the last 10 years (2009 to 2018) with the long-term normal of 1960 to 1990 (Figure 2). The highest increases in the average monthly temperature were recorded at 2.1 °C in April, 2.0 °C in February and November, 1.7 °C in August, and 1.4 °C in June and July. A statistically significant (*p* < 0.05) increase of the average temperature was noted in June (r = 0.64, *p* = 0.0001) and April (*r* = 0.34, *p* = 0.0495). In the remaining months, the increase was not statistically significant. 

Of the bioclimatic indicators, the sum of effective temperatures—growth degree days (GDD)—was assessed. The highest GDD was recorded in 2018—1646.7 degree days; the lowest was recorded in 1991—1155.9 (Appendix A). The average GDD for the period of 1985 to 2018 was 1360 degree days. The tendency of GDD increase is statistically significant with a correlation coefficient r = 0.55 and *p* value 0.0008.

The second bioclimatic indicator assessed was the Huglin index (HI). Over the observed period of 1985 to 2018, its average value was 2017. The highest HI value was 2477.5 in 2018; the lowest was 1705 in 1991 (Appendix A). The tendency of HI increase is statistically significant with a correlation coefficient r = 0.63 and *p* value 0.0001 (Figure 3).

### 2.2. Onset of Phenophases

The average onset of budburst (BBCH 08) in Welschriesling in the assessed period of 1985 to 2018 was April 20 (110 JD). The earliest date of budburst was April 20 in 2014 and the latest was May 3 in 1987 (Appendix A). The average date of Pinot Blanc budburst was April 15 (105 JD). The earliest date of budburst was March 31 in 2017 and the latest was April 28 in 1997 (Appendix A). The trend of earlier budburst in Welschriesling is statistically significant (*p* = 0.0007) with a correlation coefficient value r = −0.55. In Pinot Blanc, the trend of earlier budburst is not statistically significant (*p* = 0.07, r = −0.3). When comparing the last 10 years (2009 to 2018) with the first 10 years assessed (1985 to 1994), budburst in Welschriesling and Pinot Blanc are onset by seven and five days earlier, respectively (Figure 4).

The average monthly temperature in March has the biggest impact on budburst beginning in both varieties. The effect of a higher average temperature in March on earlier budburst beginning is statistically significant (*p* = 0.0000) and correlation coefficient values *r* = −0.74 (Welschriesling) and *r* = −0.73 (Pinot Blanc) indicate a moderately strong relationship between the variables. 

The beginning of flowering (BBCH 61): the Welschriesling variety began to flower on June 6 (161 JD), on average. The earliest date of the beginning of flowering was May 23 in 2018 and the latest was June 27 in 1991 (Appendix A). The average date of the beginning of flowering in Pinot Blanc was June 4 (155 JD), with the earliest beginning of flowering on May 5 in 2018 and the latest on June 20 in 1991 (Appendix A). The trend of earlier flowering date is statistically significant in both varieties (Welschriesling *r* = −0.51, Pinot Blanc *r* = −0.42) at *p* level < 0.05 (Welschriesling *p* = 0.002, Pinot Blanc *p* = 0.0125). Comparing to the 1985 to 1994 period, the beginning of flowering in Welschriesling was 10 days earlier and in Pinot Blanc seven days earlier on average in the 2009 to 2018 period (Figure 5).

The average maximum temperature and average monthly temperature in May have the highest impact on the beginning of flowering. With the average maximum temperature, the correlation coefficients were r = −0.82 in Welschriesling and r = −0.89 in Pinot Blanc, which indicates a moderately strong relationship between the variables. Statistically significant correlation is confirmed with *p* = 0.0000 in both varieties. The correlation coefficients for the relationship between the beginning of flowering and the average monthly temperature in May were r = −0.72 (Welschriesling) and r = −0.83 (Pinot Blanc), *p* = 0.0000 for both varieties. 

A moderately close relationship between the beginning of flowering and the average monthly temperature in April and the average maximum temperature in April is also statistically significant. With the average maximum temperature in April, the correlation coefficients were r = −0.64 (Welschriesling) and r = −0.63 (Pinot Blanc) at *p* 0.0000 and 0.0001, respectively. For the average monthly temperature in April, the correlation coefficient was r = −0.56 in both varieties (*p* = 0.0005 and 0.0006, respectively). 

End of flowering (BBCH 69): the onset of the stage corresponds to the length of flowering phenophase and the trend replicates the beginning of flowering. The average length of flowering in Welschriesling was 11 days (within the range of 6 to 15 days) while it was 12 days in Pinot Blanc (range 8 to 17 days).

Softening of berries (BBCH 85): The berries of Welschriesling began to soften on August 19 (231 JD) on average. The earliest date of the beginning of berry softening was July 28 in 2018 and the latest was September 9 in 1985 (Appendix A). The berries of Pinot Blanc began to soften on August 10 (222 JD) on average. The earliest date of the beginning of berry softening was July 20 in 2018 and the latest was September 2 in 1985 (Appendix A). In both varieties assessed, it was the last and the first year of the evaluated period. The trend of earlier softening over the assessed period of 34 years is statistically significant with *p* = 0.0000 and correlation coefficient r = −0.7 for both varieties. In the 2009 to 2018 period, the beginning of softening was 18 days earlier compared to the 1985 to 1994 period, in both varieties (Figure 6).

The highest statistically significant impact of the average maximum temperature on the date of softening of berries was found to be in June. Values of r = −0.79 (Welschriesling) and *r* = −0.76 (Pinot Blanc) indicate a moderately close relationship, *p* = 0.0000 for both varieties. A statistically significant moderately strong correlation was confirmed between the average monthly temperature in June and the softening of berries, r = −0.76 (Welschriesling) and r = −0.71 (Pinot Blanc), *p* = 0.0000 for both varieties.

A lower level of correlation, though still statistically significant at *p* < 0.05, was found between the softening of berries and the average maximum temperature in July (r = −0.4 for Welschriesling and r = −0.36 for Pinot Blanc, *p* = 0.019 and 0.037, respectively), between the softening of berries and the average maximum temperature in August (r = −0.4 for Welschriesling and r = −0.41 for Pinot Blanc, *p* = 0.02 and 0.016, respectively) and between the softening of berries and the average monthly temperature in August (r = −0.37 for Welschriesling and r = −0.4 for Pinot Blanc, *p* = 0.03 and 0.02, respectively).

Ripeness (harvesting) BBCH 89: the average harvest date of Welschriesling was October 10 (283 JD). The earliest harvest date was September 16 in 2002 and the latest was October 27 in 2008 (Appendix A). Compared to the 1985 to 1994 period, the harvest date in the 2009 to 2018 period was eight days earlier; the trend of earlier harvest date is not statistically significant (r = −0.27, *p* = 0.12).

The average harvest date of Pinot Blanc was September 30 (273 JD). The earliest harvest date was September 16 in 2015 and the latest was October 18 in 2016 (Appendix A). Compared to the 1985 to 1994 period, the harvest date in the 2009 to 2018 period was 10 days earlier; the trend of earlier harvest date is statistically significant (r = −0.4, *p* = 0.02) (Figure 7).

The strongest correlation in Welschriesling was found between the harvest date and the average monthly temperature in August (r = −0.55, *p* = 0.0008). In Pinot Blanc, the correlation between the harvest date and the average temperature in August had r = −0.4 and *p* = 0.02. Both varieties show a relatively balanced correlation of harvest date to the average monthly temperature and the average maximum temperature in June. Welschriesling showed a statistically significant correlation between the harvest date and the average temperature in June with r = −0.43 and *p* = 0.01. In Pinot Blanc, the values *r* = −0.47 and *p* = 0.005 were obtained in the correlation of the harvest date to the average monthly temperature in June, while *r* = −0.44 and *p* = 0.009 for the average maximum temperature in June. A statistically significant correlation between the harvest date and temperature in July and September was not found.

### 2.3. Interphase Intervals

Budburst to the beginning of flowering (BBCH 08 to BBCH 61): the average length of the interphase interval was 51 days in Welschriesling and 50 days in Pinot Blanc. The longest interval was recorded in 1991—71 and 70 days, respectively—while the shortest in was 1993—35 and 34 days, respectively (Appendix A). The length of the interval budburst to the beginning of flowering does not show a statistically significant trend of shortening over the observed 1985 to 2018 period in either of the varieties. 

The average GDD value for the interphase interval budburst to the beginning of flowering was 281.2 (range from 226.5 to 324.5) in Welschriesling and 242.6 (range from 194.5 to 314.4) in Pinot Blanc (Appendix A). Only in Pinot Blanc was the trend of slightly decreasing GDD statistically significant (r = −0.43, *p* = 0.012).

For the average length of the interval beginning of flowering to the end of flowering (BBCH 61 to BBCH 69) of 11 days (Welschriesling) and 12 days (Pinot Blanc), there was no significant change. The average GDDs for the 1985 to 2018 period were 99.0 (range 62.0 to 154.2) for Welschriesling and 100.1 (range 63.3 to 156.3) for Pinot Blanc. Only the trend of GDD increase in Pinot Blanc is statistically significant (r = 0.53, *p* = 0.0011).

End of flowering to berry softening (BBCH 69 to BBCH 85): the average length of the interphase interval end of flowering to berry softening over the observed 34-year period reached 59 days for Welschriesling (range 37 to 70 days) (Appendix A). The trend of shortening the length of the interval is relatively slight (r = −0.48) but statistically significant (*p* = 0.0038). When comparing the periods 1985 to 1994 and 2009 to 2018, the interval length was eight days shorter in the later period. For Pinot Blanc, the average length of the interval was 55 days (range 40 to 67 days). The trend of shortening the length of the interval is moderate (r = −0.56) and statistically significant (*p* = 0.0006). When comparing the periods 1985 to 1994 and 2009 to 2018, the interval length was nine days shorter in the later period. The average GDD value was 620.3 for Welschriesling (range 394.1 to 729.7) and 575.9 for Pinot Blanc (range 436.6 to 711). For Pinot Blanc, there is a statistically significant slight decrease in GDD value (r = −0.38, *p* = 0.0284).

Berry softening to harvest date (BBCH 85 to BBCH 89): the average length of the interphase interval berry softening to harvest date for the observed 1985 to 2018 period was 53 days for Welschriesling (range 40 to 80 days) and 51 days for Pinot Blanc (range 39 to 71 days) (Appendix A). In both varieties, there is a statistically significant moderate trend of the interval lengthening with values r = 0.51 and *p* = 0.0021 for Welschriesling and r = 0.53 and *p* = 0.0013 for Pinot Blanc. The extension of the berry softening to harvest date interval was on average nine days for Welschriesling and seven days for Pinot Blanc in 2009 to 2018 compared to the 1985 to 1994 period. For both varieties, the GDD value showed a statistically significant moderate trend of increase with r = 0.67 (Welschriesling) and 0.65 (Pinot Blanc), *p* = 0.0000 for both varieties. The average GDD value of the BBCH 89 to BBCH 89 interval was 307.3 (range 156 to 617.7) for Welschriesling and 374.2 (range 193.8 to 679.9) for Pinot Blanc.

Length of the season budburst to harvest date (BBCH08 to BBCH 89): there was no statistically significant change in the length of the season from budburst to harvest date found in either of the varieties. The average length of the season reached 173 days in Welschriesling and 168 days in Pinot Blanc, with a greater variance in Welschriesling (139 to 196 days) than in Pinot Blanc (153 to 192 days) (Appendix A). When comparing the period 2009 to 2018 and the period 1985 to 1994, the difference in the average length of the season is −1 day in Welschriesling and −5 days in Pinot Blanc. However, the r and *p* values do not confirm the trend of changing the length of the season from budburst to harvest date (r = 0.057, *p* = 0.75 for Welschriesling and r = −0.13, *p* = 0.46 for Pinot Blanc).

## 3. Discussion

The results achieved show the impact of climate change on the phenology of selected grape vine *(Vitis vinifera* L.) varieties in conditions of Slovakia (central Europe). We focused on assessing the impacts of temperature changes on grape vine *(Vitis vinifera* L.) phenology. Changes of other climatic factors have not been the subject of this work.

An increase in the average annual temperature was observed in various wine-growing regions of Europe. In south-west Germany, since the 1970s, the average annual temperatures have increased by 2.1 °C [6]; in Slovenia, there has been an increase of 0.06 °C per year [31]; in Mediterranean France, there was an increase of 1.3 °C [7] between 1980 and 2006. Our results confirm the trend of an increase in average annual temperature, with an increase of 1.4 °C indicating an increase in the average annual temperature of 0.04 °C per year.

Additionally, the average temperature of the growing season (IV to IX) shows significant changes. The increase is announced from different areas of Europe [29,31,32,33,34]. In Italy, the Venetian area experienced an increase in the average vegetation temperature (1964 to 2009) of up to 2.3°C [35]. It has reached an increase of +1.5 °C in our geographical zone, which is significantly lower than in southern Europe.

The increase of average monthly temperatures plays an important role in the earlier onset of individual phenophases. Spring thermal conditions play a key role, especially in the flowering period, which in turn affects the following phenophases [36]. Temperatures can be very variable within the wine-growing area and are closely related to the local environment [37]. In the conditions of Slovakia, the most significant increase in average monthly temperatures was recorded in February, April, and November.

From the bioclimatic indices, the GDD shows a significant increase, which was also observed in Hungary. Over the past 30 years, it has risen highly statistically above 1240 degrees [29]. In France (Burgundy), it also achieved a highly significant increase of 270 to 370 GDD [38]. In the north-west of Spain, they found a high degree of association (dependence) of the onset of phenophase and GDD [39]. In our conditions, the average GDD reached 1360 degrees with a significant growth trend and an extremely high value of 1646.7 degrees in 2018.

The Huglin index has proven to be a better predictor than other similar bioclimatic indices calculated for the growing season. This may be related to additional information contained in this index, such as the average length of the day in relation to latitude [40]. Using the Huglin index, possible extensions of areas suitable for viticulture as well as suitable grapevine varieties have been specified. The optimal grapevine varieties are substituted by those more suitable for a warmer climate (south-west Germany) [41]. The increase in the Huglin index (HI) as an indicator of the suitability of growing varieties under given conditions is confirmed in different parts of Europe [42]. In south-west Germany, it has increased from 1685 to 2063 [6]; in Hungary in the Sopron wine region, over the last 35 years the HI value has also increased above 2000 degrees [29]. The HI increase trend we have found in our conditions confirms these findings.

The findings so far confirm that northern wine-growing regions are more likely to profit from global warming. There is an anticipated earlier start of the growing season, earlier ripening of grapes and an increase in the content (quality) of grapes [33,35,43,44] with forecasts of continued acceleration of the onset of phenophases in the following decades [32].

While in the northern regions of Europe, higher temperatures in the spring period cause earlier budburst by 11 to 18 days [6,34,45], in southern Europe, Italy, and Serbia [33,35], the budburst shows great year-on-year differences, but no trend. In our terms, we found a significant slight trend of the earlier budburst of Welschriesling by seven days, but for Pinot Blanc, the trend is insignificant.

An earlier flowering in the interval of −13 to −22 days is reported by several authors [6,33,34,45] along with the fact that, unlike the budburst phenophase, there are also demonstrated trends of earlier flowering in southern Europe (in Italy for the period 1964 to 2009 by 13 to 19 days earlier flowering [35]). In our conditions, between 1985 and 2018, we found an earlier flowering by 7 to 10 days, with the most significant impact of the average and maximum temperatures in May and April. In southern regions of Europe with warmer climates, temperatures in earlier months are important; for example, in Portugal, maximum temperatures (*T_max_*) in March and April [46] are reported as significant predictors of flowering.

The advanced onset of phenophases is manifested in the beginning of berry ripening [6,33,35,45] within the interval of −13 to −22 days, which corresponds to the results achieved by us—earlier ripening in both Welschriesling and Pinot Blanc by 18 days. In our conditions, the most significant impact on this phenophase showed the average maximum and average temperature in June, followed by the average maximum temperatures in July and August, and the average temperature in August. In Portugal, *T_min_*, *T_max_* and *T_mean_* in the March to July period are reported as significant predictors of grapes ripening [46].

The greatest shift, but also a great variation, is indicated for the harvest of grapes (ripeness). Palatinate, southwestern Germany, since the 1970s has observed a harvest 25 to 40 days earlier [6]; in Italy in 1964 to 2009, a trend of 13 to 19 days earlier harvest has been observed [34]; in Mediterranean France, an earlier harvest by three weeks has been noted [7]. Earlier harvest dates are also observed in Australia [47,48]. Our results do not match with the above-mentioned data completely, because grape harvesting is more technological (winegrowing) than phenological term. In the case of suitable weather conditions, grapes are harvested later, with higher quality, which is, as mentioned above, the actual positive effect of warming (climate change) for the northern wine regions. Therefore, the late ripening Welschriesling variety does not show a significant trend of an earlier harvest date, while the Pinot Blanc variety shows a significant slight trend (−10 days).

As to interphase intervals, results from different areas of Europe vary. While in Germany (Lower Franconia) the trend of phenological intervals shortening has been observed [44], in Serbia, in the region of Sremski Karlovci, it is reported that the observed warming and change in the onset of phenophases did not significantly affect the duration of growth intervals [33]. In our conditions, we found a relatively mild, statistically significant trend of shortening the interphase end of flowering to berry softening (maturation) by eight to nine days. On the other hand, for both varieties, there is statistically moderately strong trend of lengthening the interval berry softening to harvest by seven to nine days, allowing for higher quality of the produced grape in cooler regions.

The length of the season from budburst to harvest shows great variation but there is no significant change. For the period of 1985 to 2018, Welschriesling shows an average reduction by one day, while Pinot Blanc has a reduction by five days. In Slovenia, a reduction of 15 to 27 days was reported in different varieties [31].

## 4. Material and Methods

### 4.1. Locality, Experimental Base

Long-term data obtained at the locality of the Cultivar Testing Station in Dolné Plachtince (N 48°12.327′ E 19°19.064′), which belongs to the central Slovakian wine region, district of Modrý Kameň, were used for the evaluation of climate change impact on grapevine phenology. The station is specialised in the testing of fruit crop and grapevine cultivars. The site is located in the southern part of the Krupinská planina, exposed to the south-west with 5 to 10° inclination, at the altitude of 228 m a.s.l. The soil is loam clay illimerisated brown soil, pH 6.2. The mean annual temperature (1960 to 1990 reference period) is 9.1 °C (Table 1); the average temperature of the growing season is 16.0 °C; the annual precipitation is 648 mm; the growing season precipitation is 362 mm; the average annual sunshine duration is 1983 h; the average sunshine duration within the growing season equals 1500 h. The climate of the region according to Köpper-Geiger classification is Cfb–warm temperate, fully humid with a warm summer [49]. The data were obtained from the experimental vineyard with cultivar collection liable to cultivar testing (economic and technological value of the registered, newly bred and introduced grape cultivars) planted in 3.0 × 1.2 m spacing, trained in the Rhein-Hessen form. Conventional growing technology was used during the first 20 years of the observation period, while a production system was integrated later.

### 4.2. Evaluated Varieties of Vitis vinifera L.

For the evaluation of climate change impact on grapevine phenology, Welschriesling and Pinot Blanc were chosen as model grapevine varieties. Welschriesling is the second most spread variety in Slovakia, with late ripening term, and Pinot Blanc is also among the widely grown varieties in Slovakia, with medium-late ripening.

### 4.3. Temperature Data and Indices

Temperature data of the period 1985 to 2018 were obtained from the Meteorological station in Dolné Plachtince. For the evaluation, the average annual temperature, the growing season (April to September) average temperature, the monthly average temperatures, and the average maximum temperature per month (April to September) were used.

### 4.4. Termal Indices

GDD (growing degree days)—sum of effective temperatures (effective temperature = the average daily temperature–10). A temperature of 10 °C is vegetation zero (T_base_) for grapevine (*Vitis vinifera* L.). For calculation of GDD for the locality, a period from April 1 to September 30 is considered; for the variety, a period from budburst to harvest date. 

In the work, modified GDD for principal growing season (PGS) was used, which reflects an interannual temperature variability and better expresses the impact of climate change (warming) on earlier beginning and ending of the growing season. 

PGS (principal growing season): the beginning of the PGS is the first day of the first discontinued six-day period within the year with an average daily temperature higher than 10 °C; the end of the PGS is the last day prior to the first discontinued six-day period within the year (in the autumn) with an average daily temperature lower than 10 °C.

The Huglin index (HI) is calculated using the following formula:$$\overset{Sept\;30}{\underset{April1}{\sum }}\;\;\frac{\left[\left(Tmean-10\right)+\left(Tmax-10\right)\right]}{2}\;.K$$
*T*_mean_ = daily mean temperature. *T*_max_ = daily maximum temperature. Baseline temperature = 10 °C. *K* = parameter dependent on the latitude of the location; the sum is multiplied by a factor *K* depending on the latitude of the location, taking into account the length of the day in northern latitudes (*K* for the locality of Dolné Plachtince is 1.06) [50,51].

The trends of temperature and thermal indices were evaluated with use of simple linear regression for the 1985 to 2018 period.

### 4.5. Evaluated Phenological Phases and Interphase Intervals

The BBCH scale [52] was used to set the onset of the following phenophases in grapevine (*Vitis vinifera* L.):

Budburst (BBCH 08): sprouting, green shoot tips clearly visible.

Beginning of flowering (BBCH 61): 10% of flowerhoods fallen.

End of flowering (BBCH 69): more than 80% of flowerhoods fallen.

Beginning of ripening (BBCH 85): softening of berries.

Harvest date (harvest) (BBCH 89): berries ripe for harvest.

Date of phenophase onset was transformed to Julian Day value (JD). The trends of the phenophases onset for the 1985 to 2018 period were assessed; the phenophase onset date and the average monthly temperatures were correlated.

Besides the phenophase onset, the length of the following interphase intervals (days) and the trend of interphase interval length for the 1985 to 2018 period were evaluated.

Budburst to beginning of flowering (BBCH 08 to BBCH 61).

Beginning of flowering to end of flowering (BBCH 61 to BBCH 69).

End of flowering to softening of berries (BBCH 69 to BBCH 85). 

Softening of berries to harvest date (BBCH 85 to BBCH 89).

Within each interphase interval, the average temperature of the interphase interval period (°C) and ∑ GDD of the interphase interval period was calculated.

### 4.6. Method of the Obtained Data Evaluation

The average values of the parameters, range minimum and maximum values, processing linear trends of temperature, bioclimatic indices and phenological phases and linear correlations of temperatures, bioclimatic indices and phenophase onset as well as length of interphase intervals were calculated.

### 4.7. Statistical Processing of Data

The software Statgraphics Centurion Version 18. 1. 13 (StatPoint Inc. Warrenton, VA, USA) was used for data processing-simple regression, coefficients of correlation, significance of trends and correlations at 95% significance level, *p* < 0.05 (ANOVA). Graphs were prepared with MS Excel software. 

## 5. Conclusions

Climate change has caused temperature changes in wine regions all over the world. Knowing the reaction of grapewine on these changes is important for timely adaptation of the viticultural and wine-producing sector, a proper option of varieties, and adaptation of growing technologies to changed conditions. The influence of warming on grapevine (*Vitis vinifera* L.) phenology was evaluated in conditions of central Europe, Slovakia, and Slovakian wine region based on a 34-year period (1985 to 2018) climatic and phenological data influence of temperature changes on the onset of phenophases and length of interphase intervals in Welchriesling and Pinot Blanc varieties. An increase in the average year temperature of +1.4 °C, and the average season temperature of +1.5 °C was found. The average value of HI at the locality of Dolné Plachtince in the 2009 to 2018 period was 2189, which is lower than in the 1961 to 1990 reference period—1790—which indicates an increase of HI value by 122 every 10 years. The HI value of 1790 corresponds to the cool wine region (interval 1500 to 1800) while 2189 classifies the moderately warm wine region category suitable also for very late varieties. In the evaluated phenophases, an advancement was found: earlier budburst by five to seven days, earlier beginning of flowering by 7 to 10 days, and earlier berry softening by 18 days. The obtained results confirm the fact that climate warming in central Europe (and in Slovakia) and northern wine regions has not caused changes in the grapevine phenology yet, which could call forth serious adaptation measures. Earlier veraison opens the space for increased grape quality, however.

## Figures and Tables

**Figure 1 plants-10-01020-f001:**
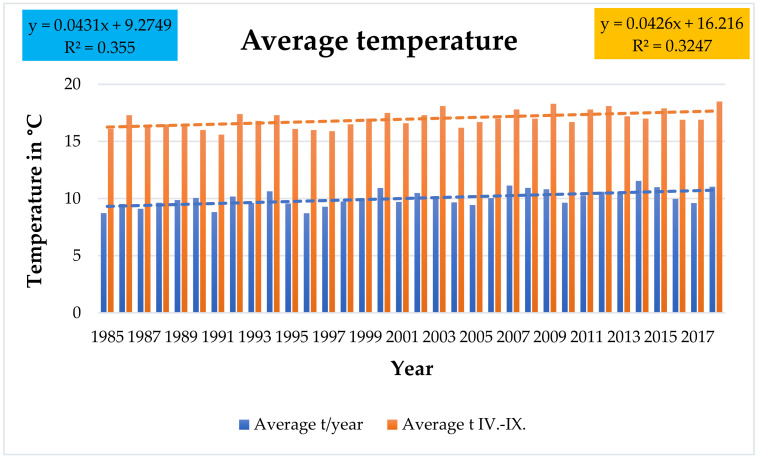
Average temperatures of year and growing season (April to September) over the 1985 to 2018 period in the experimental site.

**Figure 2 plants-10-01020-f002:**
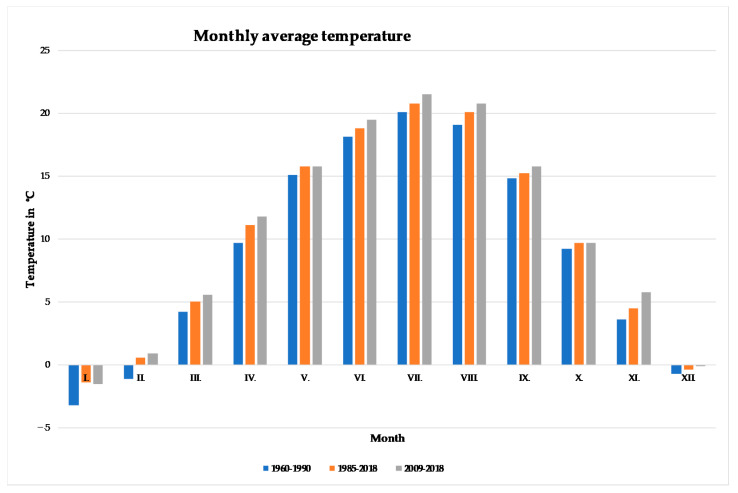
Monthly average temperatures in different periods: 1960 to 1990 (normal, reference period), 1985 to 2018 (observed period), 2009 to 2018 (last decade of the observed period).

**Figure 3 plants-10-01020-f003:**
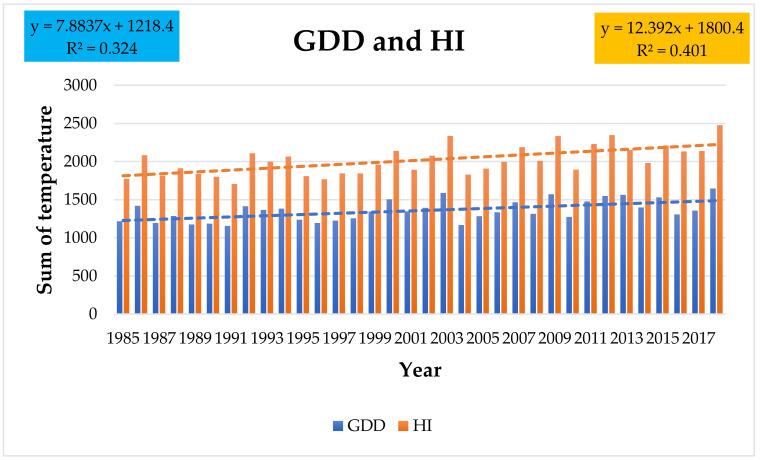
GDD and HI values in the years of the observed period.

**Figure 4 plants-10-01020-f004:**
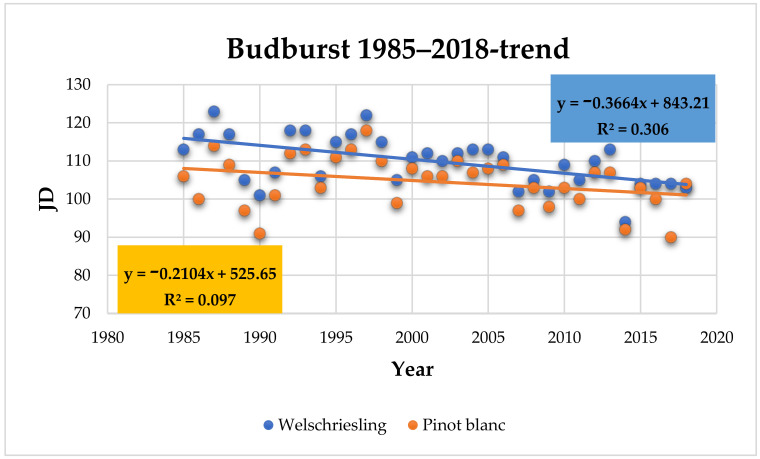
Beginning of budburst (JD) in Welschriesling and Pinot Blanc trend within the observed period.

**Figure 5 plants-10-01020-f005:**
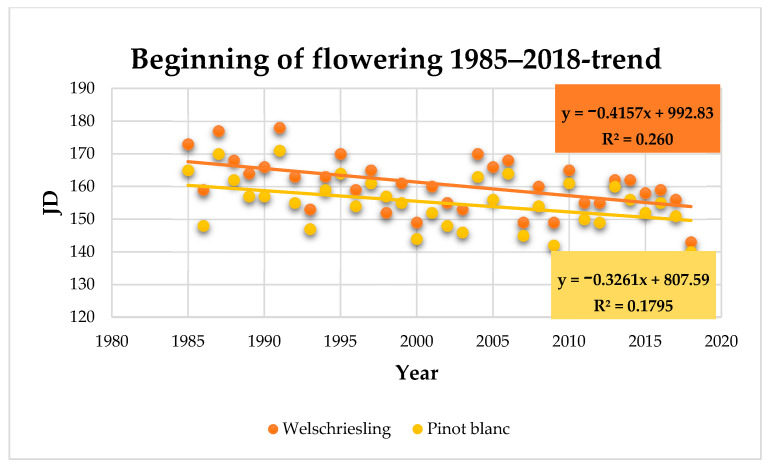
Beginning of flowering (JD) in Welschriesling and Pinot Blanc trend within the observed period.

**Figure 6 plants-10-01020-f006:**
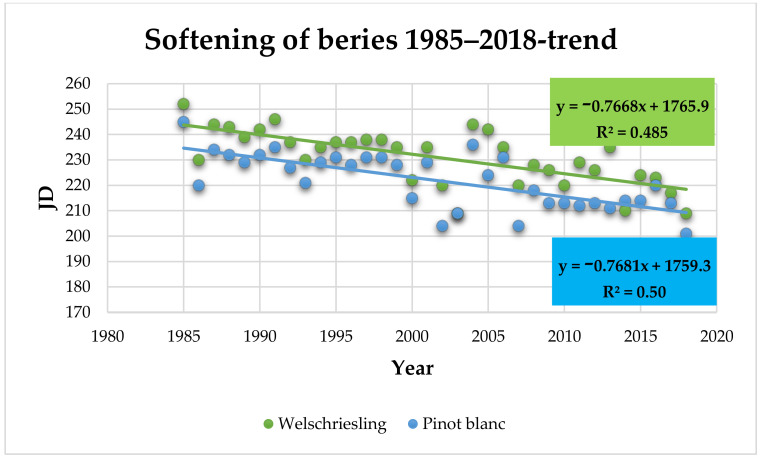
Beginning of berry softening (JD) in Welschriesling and Pinot Blanc trend within the observed period.

**Figure 7 plants-10-01020-f007:**
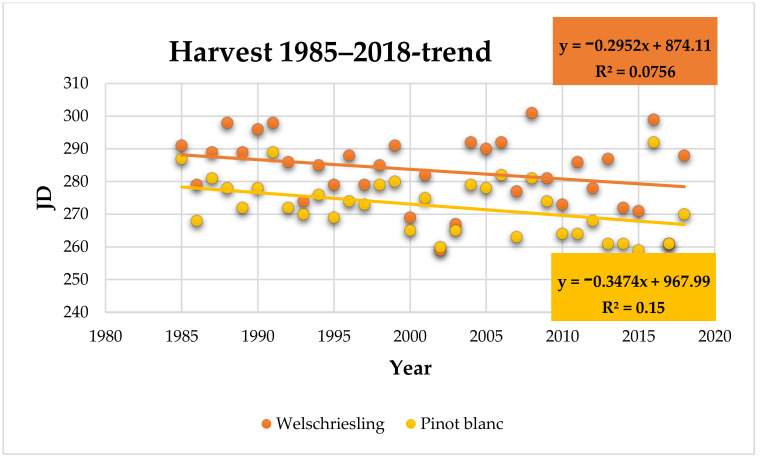
Harvest date (JD) in Welschriesling and Pinot Blanc trend within the observed period.

**Table 1 plants-10-01020-t001:** Long-term normal (1960 to 1990) of average temperatures (°C) at the locality of Dolné Plachtince.

Month	I.	II.	III.	IV.	V.	VI.	VII.	VIII.	IX.	X.	XI.	XII.	Year
Average air temperature (°C)	−3.2	−1.1	4.2	9.7	15.1	18.1	20.1	19.1	14.8	9.2	3.6	−0.7	9.1

## Data Availability

The data contained within this article are available in supplementary material to the article.

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
