# Peer review of "Influence of Climate Warming on Grapevine (Vitis vinifera L.) Phenology in Conditions of Central Europe (Slovakia)"

_plants, 2021, doi:10.3390/plants10051020_

Round 1

Reviewer 1 Report

The resubmitted manuscript on the impact of climate warming on the phenology of Welschriesling and Pinot Blanc grapevine varieties in the conditions of Central Europe (Slovakian wine region) over the period 1985-2018 has been greatly improved comparing to its previous version. Although I still consider that it lacks novelty, I agree to publish it in a Plants journal. I observed a writing mistake in Tables S6 and S7 (begininig instead of beginning).

Reviewer 2 Report

Dear Editor,

The authors of Manuscript ID plants-1202829 (Influence of Climate Warming on Grapevine (Vitis vinifera L.) Phenology in Conditions of Central Europe Slovakia)
assessed the impact of temperature change on the phenology of two grapevines varieties (Welschriesling and Pinot blanc) grown in one vineyard in the central Slovakian wine region. Specifically, they established correlations between budburst, flowering, veraison and ripeness phases with the average monthly and average monthly maximum temperatures. They also established correlations between interphase intervals and the growing degree days.

The data obtained allowed reaching some interesting conclusions about the state of viticulture in the study area in the context of global warming and whether or not there is a need for adaptation measures. Although the data is obtained from a small area of Slovakia and from only two varieties, the authors extensively related the data to previous findings in southern and northern European countries and worldwide, which highly increased the attractiveness of the manuscript.

Some minor comments:

Line 32: "have shown an average increase of growing season temperature"

Lines 118-124: from lines 388-392, four temperature indices were studied:

- average annual temperature,

- average growing season temperature,

- average monthly temperature,

- average monthly maximum temperature.

The average monthly maximum temperature is missing from the manuscript.

Line 149: “average monthly temperature” instead of “average temperature”??? The temperature indice should be clearly specified.

Line 200: is softening of berries= veraison??? The meaning of veraison should be clearly specified in the METHOD section.

Line 218: “average monthly temperature” instead of “average temperature”??? The temperature indice should be clearly specified.

Line 218: "In Welschriesling" instead of "In Pinot Blanc'. Data from Pinot Blanc have already being given earlier.

Lines 241-242: it's not clear whether the length of the interval "beginning of flowering - end of flowering" was shortened or lengthened. The information is provided for all intervals, except for this one.

Lines 285-286: "in Slovenia, there is an increase of 0.06 °C per year [46]". The reference given is for Portugal and not Slovenia. The sentence should be rewritten or deleted.

Line 326: "but no trend. In our terms..." A full stop is needed between the two sentences.

Lines 349-351: "Our results do not match with previous data". The meaning of the sentence is not clear. Which previous data? Previous data from Slovakia? Previous worldwide data? Both varieties showed a trend for earlier harvest which was significant for Pinot Blanc, and that agrees with worldwide data.

Line 377: " is 9.1°C (Table 1)??? Table 1 is not mentioned in the manuscript.

Lines 380-381: it is mentioned that data were obtained from an experimental vineyard. More information could be provided about the vineyard, it's area and how it relates to the total area of grapevine cultivation in the country, all the varieties grown on the vineyard.

Lines 424-426: In the manuscript, correlations are reported between phenophases and average monthly temperature, and between phenophases and average monthly maximum temperature. There are no correlations with GDD values and the sentence should be rewritten.

Lines 433-436: in the manuscript, correlations are reported only between the interphase intervals and the GDD. There are no correlations with the average temperature or the number of days with active temperature > 10°C. The sentence should be rewritten.

Line 456: why “2009-2018” instead of "1985-2018"?

Lines 455-458: The sentence is very confusing; 2189 is higher than 1790. Why is "reference period" in parentheses?

Lines 466-469: The supplementary materials should be cited in the manuscript.

Lines 530-531: Santos JA is not the only author for the reference.

Reviewer 3 Report

Dear Authors,

please consider the following comments.

General considerations:

This paper presents an abundant amount of data on temperature and related effects on vine phenology.  I especially found the reported trends over time very interesting.

However, the paper was somewhat difficult because the writing style needs major improvements. The authors should be more concise specially as far as Introduction is regarded. This seems to me as missing a structure. The issue about "adaptation", for instance, is repeatedly reported at lines 56, 59, 62, 75 and 83 interspersed among other issues... 

Please also explain why you exclude other relevant climate aspects such as precipitation and extreme climate events (Line 282). 

Please consider, also, the following suggestions/observations:

Lines: 43-44. The occurrence of extreme climate events you mentioned here was not considered in your study. Lines: 66-67. I completely agree with you, but these factors, namely the high genotype-specific response to climate, don't you think would have required more than the only two varieties you have chosen? Line 87: change "asi" to as it Line 134: "average" Line 141: "earliest" Line 219: "temperature" Line 375: "planina"??? Line 422: "berries" Lines 451-455-457: no divided words   1. Overall, I couldn't find in the Introduction any specific reference to the potential impact of the climate change on grapevine diseases' frequency... 4.1 Consider to report climatic classification of the study area according, for instance, to Köppen and Geiger, Thornthwaite moisture index... 4.2 A more detailed description of the agronomic characters of the adopted cultivars in the specific test conditions will be useful (e.g.: vine age, rootstock(s), applied cultural techniques, productivity, soil and plant management, phytosanitary treatments and so on...), as they can all affect and influence the vine phenology.  

Round 2

Reviewer 3 Report

Dear Authors,

even if I have noticed some attention regarding my observations, I must unfortunately observe that many of my suggestions have not actually been taken into consideration and this once again leads me to suggest that you carefully reconsider a large and articulated revision of the manuscript.

This manuscript is a resubmission of an earlier submission. The following is a list of the peer review reports and author responses from that submission.

Round 1

Reviewer 1 Report

In this study authors investigated the impact of climate warming on the phenology of Welschriesling and Pinot Blanc grapevine varieties in the conditions of Central Europe (Slovakian wine region) over the period 1985-2018.

The manuscript is well written and the data are thorougly elaborated and clearly presented, but it has a serious flaw, which is a lack of novelty. As the authors present in the manuscript, a large number of studies on this thematic have been already and systematically done in several wine growing regions all over the world, obtaining similar observations and trends as in this study. In other words, all the presented results were expected, and the authors did not manage to make an additional contribution to the existing knowledge on this topic. Possibly such study has up to now not been performed in Slovakia, but this potential fact does not increase the scientific value of the study.

Some other flaws of the manuscript are the following: the Introduction section is too extensive (too large); in the Discussion section there is a large quantity of text which does not discuss the results, but it is written like the Introduction section, giving some general information on the topic (for example lines 317-352); the obtained data are not adequately discussed, but are rather only compared to findings from other studies.

Given all the above, I do not suggest to publish the manuscript in Plants journal and authors may try to publish it in some specific technical journal concerning viticulture or climate change topics.

Reviewer 2 Report

Comments to the authors to the identification of the manuscript ID: plants-1160182 entitled " Influence of Climate Warming on Grapevine (Vitis vinifera L.) Phenology in Conditions of Central Europe (Slovakia)” Authors: Slavko Bernáth , Oleg Paulen * , Bernard Šiška , Zuzana Kusá , František Tóth

The work presented in the manuscript provides information about the impact of warming on the phenology of grapevine (Vitis vinifera L.) in conditions of Central Europe on two different grape varieties depending on their timing of ripening as a way to enable the timely adaptation of the wine sector, an appropriate choice of varieties and the adaptation of cultivation technologies to the new conditions. This type of work provides relevant information for the central European wine sector, although it is true that the results are interesting for a very localised production area.  The study is well thought out and developed in a simple way. Some standardised bioclimatic indices are included, which are a good tool as an indicator of the suitability of crop varieties in determined climatic conditions.

However, here are some aspects that should be considered by the authors:

The use of the PSD adjustment raises some doubts. What do the authors mean by discontinuous daily mean? Could they explain this modification in more detail?

This reviewer has found very few defects in form and his correction for the authors will be very fast.

Throughout the manuscript there is irregular spacing between words that the authors should correct.

 Line 120: Please, authors should replace "asl" with "a.s.l.".

Reviewer 3 Report

The objectives of the study were to assess the effects of climate on the development stages of two grapevine varieties, Pinot blanc and Welschriesling, in Slovakia. Budbreak, flowering and veraison dates were recorded at the same location from 1985 to 2018, which makes this dataset very interesting. Several indices were calculated, including the Huglin index, and statistically significant trends are highlighted. The paper is well documented.

The test used to assess the statistical significance of the trends (anova), although used in many publications, is not really appropriate. A rank test would preferable (there is no reason for searching a quantitative relationship between the “number” of the year and the value of an index).

The English grammar and style require significant improvements

Despite a nice dataset, this study only confirms many observations already made all over the world on the effect of climate change on grapevine phenology. There is no scientific novelty.

My point of view is that such a dataset should be used to test and compare different models for predicting phenological stages using air temperatures (see Morales Castillas et al for example)

Specific remarks

  • “Green tip” for budbreak is more BBCH 07 than BBCH 08 (does BBCH 08 exists for the grapevine?)
  • Why considering flowering beginning and end and not 50% of flowers open (BBCH65)?
  • The unit for the Huglin index is not “°C”
  • Line 460: results for PGS not shown?
  • Line 496: no results shown for the different parameters for the intervals between phases?
  • Line 509: what does mean “correlations at 95% level”?